# A Green Light to Switch on Genes: Revisiting Trithorax on Plants

**DOI:** 10.3390/plants12010075

**Published:** 2022-12-23

**Authors:** Diego Ornelas-Ayala, Carlos Cortés-Quiñones, José Olvera-Herrera, Berenice García-Ponce, Adriana Garay-Arroyo, Elena R. Álvarez-Buylla, Maria de la Paz Sanchez

**Affiliations:** Laboratorio de Genética Molecular, Epigenética, Desarrollo y Evolución de Plantas, Instituto de Ecología, Universidad Nacional Autónoma de México, Ciudad de Mexico 04510, Mexico

**Keywords:** epigenetics, TrxG, Arabidopsis, histone methylation, transcription

## Abstract

The Trithorax Group (TrxG) is a highly conserved multiprotein activation complex, initially defined by its antagonistic activity with the PcG repressor complex. TrxG regulates transcriptional activation by the deposition of H3K4me3 and H3K36me3 marks. According to the function and evolutionary origin, several proteins have been defined as TrxG in plants; nevertheless, little is known about their interactions and if they can form TrxG complexes. Recent evidence suggests the existence of new TrxG components as well as new interactions of some TrxG complexes that may be acting in specific tissues in plants. In this review, we bring together the latest research on the topic, exploring the interactions and roles of TrxG proteins at different developmental stages, required for the fine-tuned transcriptional activation of genes at the right time and place. Shedding light on the molecular mechanism by which TrxG is recruited and regulates transcription.

## 1. Introduction

Epigenetic regulation plays an essential role during plant development and plasticity. Plants being sessile and modular organisms require plastic responses to adjust their phenotype according to environmental conditions. The modular development of plants also implies that this process does not occur in a determined way as it does in animals; thus, organogenesis can occur throughout the plant’s lifespan. These characteristics require a fine-tuned control of gene expression, often mediated by epigenetic mechanisms [1].

This phenotypic plasticity is an emergent property of static genotypes to produce differential phenotypes in response to changing environmental conditions [2], an important phenomenon for the survival of organisms. In an increasingly changing world, the ability to generate plastic responses provides organisms with the necessary resilience for their survival; thus, understanding the mechanisms that underlie this type of response is highly relevant [3].

Epigenetics, defined as the study of mitotically or meiotically heritable changes in the control of gene expression independent of DNA sequence [4,5], shares with plasticity the peculiarity of causing changes in the phenotype (or/and gene expression) without altering the individual genome of an organism [6]. In this regard, the epigenome is largely induced by the environment, and much of the phenotypic plasticity may be governed by modifications of epigenetic marks, which module a complex integrative network between environmental changes and the adaptive capacity of plants [6,7].

Epigenetic regulation is also a key mechanism that not only controls plant plastic responses but also regulates the proliferation and differentiation processes needed for plant development. The epigenetic mechanisms modulate the homeostasis of plant meristems allowing their self-maintenance and generating cells that will differentiate into diverse cell lineages that conform to tissues or organs [8].

Among epigenetic mechanisms, post-translational histone modifications, especially histone methylations play essential roles in plant development and plasticity [9]. The Trithorax (TrxG) and Polycomb (PcG) groups are the two most important complexes that participate in almost all plant developmental processes, regulating gene expression through the deposition of H3K4me3 and H3K36me3 activation marks, and H3K27me3 repressive marks, respectively [1,10,11]. The importance of these complexes has been described through the drastic phenotypes generated by mutants (gain and loss of function) of some protein members of these complexes. In plants, the conformation, function, and mechanism have been well-identified for PcG. However, the TrxG complex has been poorly described, despite several proteins belonging to TrxG having been described [1,10,12,13,14]. So far, several proteins with features of a TrxG have been identified separately, and little is known of the TrxG complexes formed in plants and their roles along developmental processes. Therefore, in this review, we integrate and analyze recent advances mainly in *Arabidopsis thaliana* (hereafter, Arabidopsis). According to the definition of TrxG proteins, we describe new proteins with characteristics of TrxG proteins and integrate information to propose four mechanisms of plant TrxG recruitment. We also discuss the TrxG mechanism to regulate transcription, the TrxG complexes formed, as well as their roles during plant development and plasticity.

## 2. What Does It Mean to Be a TrxG Member?

The TrxG components were first identified in *Drosophila melanogaster*, as proteins whose mutations have an antagonistic function with mutations from members of the PcG complex, Trithorax (Trx) being the first protein described [15]. Despite the high degree of conservation of the TrxG factors between animals and plants, the diversification of some proteins in the different groups has originated some proteins and complexes with non-redundant and specific functions among organisms. Therefore, in plants, the TrxG factors have been identified by homology with animals, but also by their ability to: (1) suppress the PcG mutant phenotypes, (2) interact with TrxG proteins, and (3) impact on the H3K4me3 mark [1]. The composition of the TrxG complex includes heterogeneous protein types, such as histone methyltransferases (HMTs), histone demethylases, chromatin ATP-dependent remodelers, COMPASS (COMplex Proteins ASsociated with SET1) proteins, and other accessory proteins (Table 1).

### 2.1. TrxG Histone Methyltransferases

The HMTs have a SET (Suppressor of variegation 3-9 (Su(var)3-9); Enhancer of Zeste (E(z)); and Trithorax) domain, which confers HMT activity, specifically on lysine 4 (K4) and 36 (K36) from histone H3 to generate H3K4me1/2/3 and H3K36me2/3 in the case of the TrxG HMTs. Arabidopsis has more than 30 SET domain proteins in its genome, but not all of them have been characterized as HMTs related to TrxG. The TrxG HMTs have been classified mainly into three classes: (i) class II, homologous to ASH1 (Absent, Small, or Homeotic-like1), which includes ASHH1/SDG26 (SDG26), ASHH2/SDG8/EFS (SDG8), ASHR3/SDG4 (SDG4), as well as ASHH3/SDG7 and ASHH4/SDG24, two proteins whose function as TrxGs is still unknown; (ii) class III, homologous to Trx/Trr, which contains ARABIDOPSIS TRITHORAX (ATX) and ARABIDOPSIS TRITHORAX RELATED (ATXR) proteins: ATX1, ATX2, ATX3, ATX4, ATX5, ATXR3/SDG2 (SDG2), and ATXR7/SDG25 (ATXR7); and (iii) class VI that contains ATXR2/SDG36 (ATXR2), an HMT with a SET domain truncated or interrupted [46,47]. It has been reported that ATX1, ATX2, ATX3, ATX4, ATX5, ATXR2, ATXR7, SDG2, and SDG4 catalyze preferentially the H3K4me3 mark, whereas H3K4me2 is catalyzed by ATX2, ATX3, ATX4, ATX5, and the H3K36me3 by ATXR2, SDG26, SDG8, and SDG4 [16,17,18,22,24,25,26,29,31,48,49]. A dual function has been reported for ATX1, ATX2, and ATXR7, which also seem to catalyze the H3K4me1 mark deposition [24,50]. The reported specificity of these HMTs (Table 1) is based mainly on analyses of loss-of-function mutants but not on biochemical analyses, being analyses performed at a specific developmental stage, without considering their different interactions or other factors that could be necessary for specificity.

Therefore, although there are different HMTs with the same specificity, the expression pattern, and interactions among TrxG proteins allow generated diversity and specificity on different zones across the genome. For instance, a reduction of 46.4% in all H3K4me3 marked sites in the genome was observed in the *sdg2* loss of function mutants, while the *atx3 atx4 atx5* triple mutant showed only a reduction of 16.3% in H3K4me3 sites, but only 5.4% of them were shared between *sdg2* and the *atx3 atx4 atx5* triple mutant [18]. Moreover, the phenotype analyses of these mutants indicate that ATX3, ATX4, and ATX5 are redundant with ATX1 but not with ATX2 or SDG2 [18].

Moreover, the distribution of the TrxG marks is different across the genome; H3K4me2/3 accumulates predominantly on promoters and near the transcription start site (TSS) of the genes, whereas H3K4me1 is found over the transcribed region. Genome-wide analyses indicated an overlap between H3K4me2 and the PcG repressive mark H3K27me3, suggesting the existence of different roles between the different H3K4 methylation degrees [51]. The H3K36me3 mark is enriched in the 5′-half of transcriptionally active genes, while H3K36me2 has a peak at the 3′-end of the active genes [52].

### 2.2. Histone Demethylases

In addition to the deposition of H3K4me3 or H3K36me3 active chromatin marks, the TrxG complex also includes histone demethylases responsible for removing or restricting the spread of PcG repressive marks, establishing boundaries around active genes [53]. Six histone H3K27me3 demethylases have been described related to TrxG (Table 1). EARLY FLOWERING 6 (ELF6), RELATIVE OF EARLY FLOWERING 6 (REF6), and JUMONJI DOMAIN-CONTAINING PROTEIN13 (JMJ13), which have redundant roles in the reactivation of the *FLOWERING LOCUS C (FLC)* gene in embryogenesis, after vernalization [42,53,54], and JMJ26 which interacts with ATX3, as well as JMJ24 and JMJ28 that interact with the COMPASS proteins ARABIDOPSIS ASH2 RELATIVE, (ASH2R), WD40 REPEAT5a (WDR5a), and RBBP5 LIKE (RBL). In addition, JMJ24 also interacts with ATX1, ATX2, ATX4, and ATX5, and JMJ28 with ATX4 and ATX5, suggesting that these proteins could form different TrxG complexes [39].

### 2.3. Chromatin Remodeler Factors

In general, chromatin remodeler factors use the energy of ATP hydrolysis to shift, destabilize, or restructure chromatin to regulate DNA accessibility and transcription. SPLAYED (SYD), BRAHMA (BRM), PICKLE (PKL), and INO80 (INOSITOL REQUIRING 80) have been characterized as ATP-dependent chromatin-remodeling factors members of the TrxG complex [55,56,57,58,59]. BRM and SYD have been considered TrxG components for their ability to activate gene expression and suppress the PcG mutant phenotypes [58,60,61]. Likewise, the interaction of PKL with ATX1 and its antagonistic function with PcG proteins have led to it being considered as part of TrxG [57]. INO80 interacts with a COMPASS-like complex and participates in transcriptional activation by regulating the H3K4me3 mark deposition [39,59]. It is worth noting, however, that the roles of SYD, BRM, and PKL are not exclusive to TrxG, as they also act synergically with PcG [62,63,64].

### 2.4. COMPASS Proteins

The core of Trithorax complexes is constituted by COMPASS proteins. The first TrxG COMPASS complex was described together with the yeast HMT SET1; thus, any other complexes that contain HMTs different from SET1 or their homologs in other organisms are considered COMPASS-like, instead of a COMPASS complex [65]. These COMPASS proteins stimulate the activity of HMTs, mainly those that catalyze the H3K4me3 mark, being essential for their activity [66,67]. However, in Arabidopsis, HMTs are active even in the absence of COMPASS proteins, indicating that they are not essential for HMT activity [68].

In Arabidopsis, the COMPASS core consists of three proteins homologous to those in animals and yeast: ASH2R, RBL, and WDR5a [32,34]. ASH2R mediates H3K4 trimethylation and RBL is necessary for ASH2R-WDR5a interaction, whereas WDR5a is fundamental for the interaction with HMTs [34]. The overlapping spatial expression patterns of COMPASS proteins during plant development, suggest that they partner together consistently.

Other proteins not explicitly described as members of Arabidopsis COMPASS are S2lb and some JMJ proteins. S2lb, homolog of yeast WD repeat domain protein Swd2 and WD82 in mammals, is crucial for the maintenance of the H3K4me3 landscape throughout the genome; its physical interaction with the core COMPASS protein WDR5a and with the HMT SDG2 clearly defines it as a Trithorax member [36,69]. In addition, the interaction of JMJ24 and JMJ28 with WDR5a and RBL has led to suggest that JMJ proteins can act as components of the COMPASS complex [39].

Up to now, there are no reported interactions of COMPASS proteins with ATXR7, the Arabidopsis ortholog to SET1, and consequently, no TrxG COMPASS has been described in plants. Nevertheless, the interaction of COMPASS proteins with ATX1, an HMT homologous to Trr/Trx, has been reported, and thus this complex has been defined as COMPASS-like [34,70]. The interaction of ATX3 and ATX4, two H3K4me3 HMTs with WDR5a [34], as well as the interaction between SDG2, S2lb, and WDR5a, suggest the existence of other COMPASS-like complexes.

### 2.5. TrxG Accessory Proteins, the Case of ULTRAPETALA1 and SECRET AGENT

ULTRAPETALA1 (ULT1) is a plant-exclusive TrxG component found only in angiosperms [71], which contains a SAND-domain protein, and has been characterized as a TrxG component by its physical interaction with ATX1 and antagonistic function with PcG [72]. Despite lacking an enzymatic domain in its structure, ULT1 has a positive effect on the H3K4me3 mark, apparently by stimulating ATX1 activity, a feature that has led to its description as an ATX1 co-activator [72]. The ULT1 function together with ATX1 seems to be restricted to some aerial tissues given that in roots, ULT1 acts independently of ATX1 [71,73].

Besides ULT1 function, recent evidence postulates the existence of another protein that stimulates ATX1 activity. The protein SECRET AGENT (SEC), an O-linked β-N-acetyl-glucosamine (O-GlcNAc) transferase, enhances ATX1 activity through O-GlcNAcylation modification on the Ser947 of its SET domain, acting like an enzymatic coactivator [74]. The GlcNAcylation is a common modification involved in many other processes [75], hence SEC could have many targets, including non-epigenetic proteins, so it could be considered as a common modifier influencing ATX1 activity. However, SEC fulfills several characteristics of TrxG proteins: its interaction with ATX1, which allows it to stimulate the deposition of H3K4me3, and in turn, counteract the effect of the PcG mark [74]. Moreover, in mammals, the integrity of the SET1-COMPASS complex is regulated by O-GlcNAcylation [76]. Therefore, SEC could be considered as part of TrxG, evidencing a regulation mediated by GlcNAcylation conserved in plants.

## 3. A Call for TrxG to Activate Genes

The mechanisms of TrxG recruitment into DNA have been described mainly in *Drosophila* and mammals, where TrxG machinery is recruited at least in three different manners by (i) sequence-specific DNA binding, (ii) the recognition of pre-existing histone modifications, or (iii) long non-coding RNAs (lncRNAs) [65,77]. Although in plants, TrxG recruitment is not well described, there is evidence that the mechanism of recruitment by sequence-specific DNA binding, the recognition of pre-existing histone modifications, and lncRNAs are conserved in plants (Figure 1).

Sequence-specific DNA binding occurs by a special class of Cis-regulatory elements known as Trithorax response elements (TREs) which are enriched in transcription factor binding sites (TFBSs) and often coincide with PcG response elements (PREs) [65,78]. In analogy to the recruitment of PcG to CpG Islands (CGIs), it has been suggested that in mammals, COMPASS-like complexes can bind CGIs through the CxxC domain of HMTs, MLL1, and MLL2 [65]. In Arabidopsis, the CxxC domain is not conserved in TrxG proteins. In addition, there is no evidence of the direct binding of TrxG components at sequence-specific DNA, nor for CGI involvement in the recruitment of PcG or TrxG complexes. Nevertheless, in *Oryza sativa*, OsULT1 binds to DNA GAGAG motifs of the cold stress response *OsDREB1b* gene, where its SAND domain seems to play an important role in its specificity [45]. The GAGAG motif is one of the cis motifs present in both plants and *Drosophila* PREs [79,80,81,82]. The interaction of OsULT1 with OsTrx1, which in turn interacts with OsWDR5a [33], suggests a conserved mechanism of the TrxG recruitment sequence specific through OsULT1 in plants (Figure 1A).

One example of TrxG recruitment associated with TFBSs is found in the *FLC* regulation, a transcription factor (TF) involved in flower transition during vernalization [83]. The cis-acting cold memory element (CME) found within the *FLC* large intron, contains two RY motifs described for their importance in both PcG and TrxG recruitment [84,85,86,87,88]. RY motifs are recognized by TF with B3-domains, specifically, VAL1 and VAL2, which are able to recruit PcG proteins to repress *FLC* expression; in the same way, LEC2, FUS3, and ABI3 recruit TrxG proteins to establish an active chromatin state at *FLC* [86]. In this recruitment, LEC1 partly promotes *LEC2*, *FUS3*, and *ABI3* gene expression and binds to the *FLC* distal promoter region through its B3 domain to promote an open chromatin state and increase the chromatin accessibility of the CME [86,87,88]. Upon expression, LEC2, FUS3, and ABI3 bind to the RY motifs and act in partial redundancy to displace PcG factors, and in turn, recruit the FRI (a scaffold protein) complexed with TrxG factors such as COMPASS-like proteins and H3K36 methyltransferase SDG8 to activate *FLC* expression [86] (Figure 1B).

TrxG recruitment by the recognition of histone modifications has been documented in plants [89]. In Arabidopsis, the monoubiquitination of lysine 143 of histone H2B (H2BK143Ub1) is increased concomitantly with the H3K4me3 mark, and recent information indicated that H2Bub correlates with the localization of ATX1 and ATX2 into the genome [50,90]. Indeed, the regulation of *FLC* by ATX1 requires H2Bub1 to maintain an active transcription state [91] (Figure 1C). This recruitment by H2Bub1 seems to be specific for some HMTs. The H3K4me3 deposition mediated by SDG2 together with S2Lb is independent of H2Bub1 [36]. Although Swd2, the homolog in yeast to S2Lb, is capable of recognizing H2Bub1 to promote H3K4me3 deposition [92], S2Lb does not seem to maintain this function [36]. This information reveals that the recognition of TrxG proteins by H2Bub1 is not a widespread mechanism.

Even though plant ncRNAs are most associated with transcriptional repression, there are some examples associated with TrxG. Such is the case of *MADS AFFECTING FLOWER* (*MAF4*) antisense lncRNA called *MAS* from Arabidopsis and *LAIR* from rice [93,94]. *MAF4* is another TF closely related to *FLC*, where the antisense lncRNA *MAS* is transcribed, from its TTS (Terminal Transcriptional Site) into the first intron of *MAF4* [94]. Similar to lncRNAs HOTTIP, NeST, and Evx1as in humans [95,96,97], *MAS* has been found to bind WDR5a and guide the COMPASS-like complexes to *MAF4* locus to promote gene activation through H3K4me3 deposition [94] (Figure 1D).

This type of TrxG recruitment is conserved in monocots as well, specifically in rice. *LAIR* is an antisense lncRNA transcribed from the 5′ terminal region of the *LRK* gene cluster, found to be involved in grain yield [93]. The silencing of *LAIR* causes *LRK* repression, while the overexpression of *LAIR* increases the expression of some *LRK* genes [93]. This is possible due to *LAIR*’s ability to interact with OsWDR5a and carry the TrxG machinery to promoters and activate transcription [93].

A set of lncRNAs also play an important role in *FLC* regulation [98]. *COLDAIR*, a sense lncRNA found within *FLC* long intron, participates in *FLC* repression and allows transition to flowering when vernalization conditions are present [99]. However, *COLDAIR* overexpression seems to have the opposite effect on *FLC* expression, maintaining *FLC* activation, associated with an increase in H3K4me3 marks and H3K27me3 reduction [100]. It may be tempting to conclude that *COLDAIR* may have a dual role in regulating *FLC* expression both ways, and thus be a “natural” mechanism to activate its expression [100]; however, we cannot rule out that the high levels of H3K4me3 could be due to the PcG machinery being hijacked by ectopically expressed *COLDAIR* transcripts. Although it is not a direct example of TrxG recruitment, it is important to point out this *COLDAIR* role for future investigations.

While we can describe only a few known examples of TrxG recruitment in plants, we must consider the different rates at which discoveries are being made regarding PcG proteins compared to advances made in animals. Since several years ago, we have known the number of lncRNAs present in mammals as well as their interactions with members of the PcG complex PRC2 [101]. Not only that but we also know the pathway by which WDR5a binds lncRNAs to regulate its target genes properly and maintain homeostasis in embryonic stem cells [102]. However, only a couple of years ago, the first massive search for lncRNAs was undertaken in Arabidopsis [94], which already means a big breakthrough towards understanding mechanisms that call TrxG proteins to action. Therefore, so far, there are examples of the recruitment of TrxG individual proteins, but it will be necessary to understand their interactions and recruitment through the basal transcriptional machinery.

## 4. How Does TrxG Machinery Activate Transcription?

While we mention the histone modifications established by TrxG machinery as transcriptional active marks, we must ask ourselves what their role in gene transcription is. In this regard, an interplay between the ATX1-COMPASS-like complex and transcription machinery has been reported in plants [70].

Gene transcription consists of a series of well-described steps. In the first phase, the preinitiation complex assembly requires ATX1-COMPASS-like proteins, where ATX1 binds directly to non-phosphorylates CTD of Pol II; for this step, the ATX1 activity is dispensable. After preinitiation complex assembly and the Ser5P phosphorylation of CTD-Pol II, Pol II moves to the TSS to transit to the elongation phase of transcription. At this step, Ser5P Pol II recruits ATX1; in turn, ATX1 recruits the COMPASS complex to induce the deposition of H3K4me3, a mark required to transit to the elongation phase of transcription [70,103]. This mechanism implicates a dual role of ATX1-COMPASS-like in transcriptional activation; in an inactive form, it contributes to the assembly of the basal transcription machinery, whereas in an active form, it facilitates the transition of Pol II into the elongation phase [103].

Although this mechanism includes some important aspects of the link of TrxG with transcription machinery, further experiments are needed to understand this mechanism in more detail as well as if this is a generalized mechanism for other plant TrxG complexes.

## 5. Trithorax Teaming Up: Known TrxG Complexes

Unlike PcG, where specific PRC2 subcomplexes have been identified to act in a development stage-dependent manner [14], TrxG complexes that participate in the different plant developmental stages are not clearly defined. Furthermore, most TrxG complexes have been described for the first time by their role in *FLC* regulation.

Indeed, the first plant TrxG complex described, hence the most widely known, is an ATX1-COMPASS-like complex containing WDR5a, RBL, and ASH2R that regulates *FLC* expression [34]. Moreover, the interaction between ATX1 and CBP20, a subunit of the CAP BINDING COMPLEX (CBC) involved in mRNA 5′ cap binding and pre-mRNA splicing, has been reported, as well as the ATX1 genetic interaction with ASH2R and SDG8 that participates in *FLC* regulation [68]. CBP20 is required for the deposition of both H3K4me3 and H3K36me3 marks, and reciprocally, SDG8, ATX1, and ASH2R are necessary for the CBC-mediated mRNA cap protection and efficient pre-mRNA splicing [68]. Therefore, we can predict a TrxG complex containing HMTs with H3K4me3 and H3K36me3 activity, COMPASS proteins, and elements of mRNA metabolism that participate in the regulation of floral transition through *FLC* regulation (Figure 2A). Additionally, an ATX1-COMPASS-like complex containing WDR5a, RBL, and ASH2R that regulates other genes related to biotic stress has been reported [70].

Another TrxG complex that also participates in floral transition and other processes of development, is formed between INO80 and COMPASS-like proteins [39,104]. This complex includes the three core COMPASS proteins WDR5a, RBL, and ASH2R, and the HMTs ATX4, ATX5, as well as JMJ24 that acts as a bridge to assemble INO80 with the rest of the complex [39]. Within this complex, various accessory subunits of INO80 were also identified as taking part in the interaction: INB1, INB2A,2B; INB3; NFRKB 1,2,3; UCH1,2; YY1; ARP4; ARP9; IES2A,B,C; RIN1,2; ARP5; and EEN, most of them essential for INO80 catalytic activity [39] (Figure 2A).

The only characterized complex that includes an HMT that does not belong to the ATX1-5 group is formed by SDG2, S2lb, and WDR5a, which seems to be an important complex for the global maintenance of the H3K4me3 mark, participating in diverse developmental processes [22,36,105] (Figure 2B). Both S2lb and SDG2 are major contributors to the H3K4me3 mark throughout the genome, and their pleiotropic phenotypes of loss of function mutants suggest their participation in different developmental processes [22,36,105]. Although in this complex, the presence of the other two member proteins of the COMPASS core (ASH2R and RBL) is just hypothesized, the interaction with CDKC;1, a protein that mediates RNA Pol II CTD Ser-2 phosphorylation, integrates a link towards transcriptional elongation [36,106].

The interaction between OsWDR5a and OsTRX1/SDG723, an HMT of H3K4 from *Oryza sativa*, predicts the existence of a TrxG complex in rice that regulates flowering by the deposition of the H3K4me3 mark at the *Ehd1* locus, as well as rice panicle development [107,108].

As the characterization of some of these TrxG complexes has been performed in a specific locus or developmental process, it is unknown if they participate in other processes or if their function is restricted to a certain process or tissue. Therefore, it is important to be careful in generalizing the involvement of these complexes to other developmental processes. For instance, ULT1 was first described as a TrxG member, for its physical and genetic interaction with ATX1 in the inflorescence meristem, where ATX1 together with ULT1 regulates the expression of a subset of the floral genes [72]. However, genetic interactions indicated that in the Root Apical Meristem (RAM) the ULT1 roles are independent of ATX1 [71] (Figure 2C). This functional independence between ATX1 and ULT1 in the root not only reveals the versatility of ULT1 but also predicts the existence of different tissue-dependent TrxG complexes which contain ULT1 interacting with different HMTs other than ATX1.

## 6. Role of TrxG Proteins during Development

Historically, most of the functions of different TrxG proteins have been reported by studies of the loss and gain of function mutants at specific developmental stages or on specific targets of PcG. In order to understand the implications of the different TrxG members during plant development, we summarize the information regarding the roles of the different proteins of the TrxG complex during vegetative growth, flowering transition (Figure 3A), and root development (Figure 3B).

### 6.1. From Vegetative Growth to Flower Development

ATX1 regulates the flowering time and flower development through the activation of the homeotic TF *APETALA2 (AP2)*, *PISTILATA* (*PI)*, and *AGAMOUS (AG)* [109]. ATX1 and ATX2 act redundantly in *FLC* regulation, but only ATX1 binding on the *FLC* gene has been reported [16]. ATX3, ATX4, and ATX5 have redundant roles in vegetative and reproductive development, certainly, the *atx3 atx4 atx5* triple mutant is a dwarf plant with smaller leaves and fewer seeds but with a normal flowering time [18]. ATX4 and ATX5 also participate in flowering time in long-day conditions [39]. Meanwhile, SDG2, the principal contributor of H3K4me3 deposition in Arabidopsis, plays a crucial role in vegetative and reproductive development. SDG2 regulates flowering time through *FLC* activation and has important roles in male and female gametogenesis; in fact, the *sdg2* mutants are sterile [22,105]. ATXR7 also regulates flowering time through direct *FLC* activation; however, this regulation seems to be independent of that of ATX1 [24]. SDG8 also directly regulates *FLC* expression by the deposition of H3K36me3, whereas it also interacts with the demethylase ELF6 to reduce the H3K27me3 mark [42]. SDG26 promotes flowering time through the direct activation of the TF *SUPPRESSOR OF OVEREXPRESSION OF CONSTANS 1 (SOC1)* by the deposition of the H3K4me3 and H3K36me3 marks and, in contrast, other HMTs such as ATX1, ATXR7, SDG2, and SDG8; the *sdg26* loss of function mutant has a late flowering phenotype [29]. SDG4 interacts with TF AMS (ABORTED MICROSPORES) to regulate stamen development [110] and has also been shown to control pollen tube growth, affecting the fertility of the loss of function mutants [31].

As mentioned above, the COMPASS core proteins, ASHR2, RBL, and WDR5a, also regulate the flowering transition through *FLC* regulation [32,34], whereas S2Lb is also important to promote flowering development, by an unknown mechanism.

Moreover, BRM regulates the flowering transition by the direct activation of the *SHORT VEGETATIVE PHASE (SVP)* flowering repressor, inhibiting the PcG recruitment at that locus [60]. SYD regulates the Shoot Apical Meristem (SAM) identity by the direct activation of *WUSCHEL (WUS*), the principal gene maintaining the Stem Cell Niche (SCN) [38]. SYD and BRM also regulate flower determination by interacting with the TFs SEPALLATA3 (SEP3) and LEAFY (LFY), which contribute to the recruitment of these chromatin remodeling factors to activate the transcription of *AG* and *APETALA3* (*AP3)* flowering identity genes [58]. Furthermore, SYD and BRM share common targets involved in other developmental processes, e.g., some genes of the auxin pathway [63]. Indeed, it has been reported that MONOPTEROS (MP/ARF5), an auxin response factor, recruits SYD and BRM to increase chromatin accessibility [111].

PKL regulates the flowering time by activating the *FLOWERING LOCUS T* (*FT)* gene, promoting H3K4me3 deposition together with ATX1, and preventing H3K27me3 deposition. In this regulation, CONSTANS (CO) mediates PKL recruitment at the *FT* locus [57,112].

Histone demethylases also participate in flowering. For instance, REF6, ELF6, and JMJ13 have redundant roles promoting the flowering transition, and the phenotype of a triple mutant shows pleiotropic phenotypes, including dwarf leaves, smaller siliques, and some severe defects such as a loss of floral determinacy and flowers with more petals [53]. These phenotypes indicate that the histone demethylases are involved not only in the flowering time but also in the meristem maintenance and the flower identity. In addition, JMJ24 also promotes flowering transition, apparently together with COMPASS proteins and ATX4 and ATX5 [39].

ULT1 regulates flower development by regulating *AG* and *AP3* expression. ULT1 interacts with ATX1 to deposit the H3K4me3 mark at the *AG* locus and prevents PcG H3K27me3 deposition [72]. ULT1 also regulates the SAM size by limiting *WUS* expression. Indeed, it has been suggested that floral meristem enlargement is responsible for the generation of extra floral organs in *ult1* mutants [72,113].

### 6.2. Root Development

ATX1 plays an important role in root development; it controls primary root growth by regulating meristematic cell proliferation in the RAM and participates in lateral root emergence. In the root SCN, ATX1 is necessary to maintain lower levels of Quiescent Center (QC) cell division, these being auxin-independent phenotypes [71,114]. SDG2, another HMT, also participates in promoting primary root growth, lateral root formation, the maintenance of the RAM cell number and size, as well as the maintenance of SCN organization and an undifferentiated state of columella stem cells. Contrary to *atx1* mutants, *sdg2* phenotypes are associated with a diminished auxin response [115], suggesting that their roles in root development are by different pathways. This corresponds to the non-redundancy between SDG2 and ATX1 in aerial tissues [18]. The lower levels of *ATX3*, *ATX4*, and *ATX5* expression reveal no relevant function in root tissues [18]. SDG4 is another HMT involved in the root SCN maintenance, controlling the columella stem cell differentiation and the QC cell identity by the deposition of the H3K4me3 mark at the promoter of *WUSCHEL RELATED HOMEOBOX 5* (*WOX5)*, a QC cell identity gene [49]. ATXR7 is also necessary to maintain an undifferentiated state of columella stem cells but is less penetrating than SDG4 [49]. It has been shown that SDG26 is expressed in cortical root cells [116], but its function in root development remains unknown.

S2Lb is the only reported COMPASS protein involved in root development, promoting primary root growth and lateral root formation [36].

The ATP-dependent chromatin remodeling factors have pleiotropic functions including root growth and RAM maintenance. BRM is preferentially expressed in meristematic tissues, and in roots, it promotes primary root growth, meristematic cells proliferation, and epidermal cell elongation, and is also involved in maintaining QC cell identity and the undifferentiated state of columella stem cells. BRM controls root auxin response by regulating the expression of *PLETHORA 1* (*PLT1)* and *PLETHORA 2 (PLT2)*, two TFs involved in the auxin pathway, and also by directly inhibiting H3K27me3 deposition in the auxin-transporters *PIN-FORMED 1* (*PIN1)* and *PIN-FORMED 2* (*PIN2)* [117]. In contrast to aerial tissues, SYD seems to play no role in the root [118]. The subunit, INO80, and its subunit, ARP5, promote primary root growth by stimulating the meristematic cell proliferation of the RAM [119].

PKL is expressed in the RAM including the SCN and promotes primary root growth and meristematic cell proliferation, maintains the columella stem cell undifferentiated state, and activates some QC cell identity genes [40].

The demethylase REF6 is also expressed in roots and is necessary for lateral root formation by regulating *PIN1*, *PIN2*, *PIN3*, *PIN4*, and *PIN7* expression and counteracting the H3K27me3 deposition of those genes [120].

ULT1 participates in the root SCN homeostasis and maintains an adequate QC cell division rate and an undifferentiated state of columella stem cells. Moreover, ULT1 promotes the expression of a subset of auxin-related genes and maintains proper auxin signaling at the root tip [71]. Although ULT1 can interact physically with ATX1 and both have similar functions in SAM differentiation, the ULT1 roles in the root SCN are ATX1-independent [71].

## 7. Life Being Plastic, Is Fantastic! Trithorax and Plant Plasticity

One of the many reasons why epigenetic regulation is such a hot topic in scientific research is that it is often viewed as an area capable of bridging the gap between the intrinsic processes of organisms and their environment; thus, it is fair to say that, at least in some extent, this is true. As mentioned at the beginning, TrxG is one of the players mediating this connection between what is happening within the plant and what is in the outer environment, important for plant survival by giving them the ability to have plastic responses.

A very good example of phenotypic plasticity in plants is the phenomenon of thermomorphogenesis, where a change induced by high temperatures can generate several phenotypic variations, such as the elongation of the hypocotyl and petiole, leaf hyponasty, and even early flowering, in order to favor a more suitable phenotype for cooling down to the plant [121,122]. This set of changes is mediated by a complex molecular regulation. In this sense, some TrxG members have been identified to be involved. For instance, INO80 is able to regulate the expression of auxin and thermoresponsive genes in elevated temperatures by evicting histone variant H2A.Z, mediating H3K4me3 deposition, and associating with transcription elongation factors [59]. Both ATXR7 and ATX1 participate in activating response genes for thermotolerance involved in the initial heat-stress response and maintenance during recovery [123], providing insights into the role of TrxG during thermomorphogenesis.

WDR5a, for its part, also plays plastic roles, specifically, modulating root growth in response to toxic metals [124]. In particular, cadmium induces *WDR5a* expression, which in turn activates the NOS-like enzyme, which causes an accumulation of NO (Nitric Oxide) to repress the gene expression of the auxin pathway, and thus, root growth is arrested [124,125]. Additionally, WDR5a has an impact on another plastic response: drought tolerance, achieved by the same NO accumulation mentioned before, by regulating stomatal closure and aiding the transcription of stress-responsive genes [126].

Interestingly, drought tolerance is a phenomenon in which many TrxG components converge, the H3K4me3 mark has been identified as very changeable when plants are subjected to this type of stress [127]. In this sense, ATX1 is involved in the expression of numerous genes involved in drought tolerance through the abscisic acid (ABA) pathway and independently of it; specifically, it has been demonstrated how this HMT induces the expression of the enzyme NCED3, a key biosynthetic protein in the ABA pathway [128]. Counterintuitively, ATX4 and ATX5 have been proposed to negatively affect plant sensitivity to drought; their mutations showed enhanced drought tolerance during germination and seedling development, this through aiding the transcription of *AHG3*, a negative regulator of the ABA pathway [129].

Enzymes involved in the response to abiotic stress such as SIZ1, which post-translationally sumoylates different proteins, recognizes H3K4me3 and physically interacts with ATX1 to avoid its activity, suppressing the transcription of genes such as *WRKY70* [130]. Additionally, ULT1 has been shown to regulate lots of genes related to different abiotic stresses, its mutant displays a significant amount of differentially expressed genes belonging to categories related to oxidative, drought, chemical-induced, and cold stresses [131].

Furthermore, ULT1 was also identified as a regulator of genes related to biotic stress responses, e.g., innate immune response and fungi defenses. In fact, ULT1 is also a key regulator of enzymes of the glucosinolate pathway, which is importantly involved in plant response to herbivores and pathogens [131,132]. In responding to biotic stressors, SDG8 and SDG4 have been reported as important for facing pathogen infection, specifically *Pseudomonas syringae*; both proteins were shown to be downregulated upon infection and to present hypersensitive responses upon infection [133,134]. Indeed, SDG8 is important for the sustained transcription of key defense genes, *PR1* and *PR2* [134].

In some stress processes, plants can generate an epigenetic stress memory, in which preexposure to biotic or abiotic factors may induce a memory that enables a faster or stronger response to subsequent stress [135]. In this process, the H3K4me3 mark deposited at some responsive genes in the first stress exposition is maintained allowing a faster and, in some cases, stronger gene expression in a second stress. In this memory process, ATX1 and SDG8 participate in the deposition of histone marks [136,137].

## 8. Trithorax Breaking the Mold: Atypical Roles in the Regulation

As has been mentioned throughout this review, the TrxG has historically been analyzed in a very stereotypical way. The regulation of its targets has usually been thought to be shared and antagonistic to those of the PcG, exclusively participating in transcriptional activation. However, it is important to point out that there is increasing evidence that PcG and TrxG proteins can participate in contrary processes to their canonical function; in other words, TrxG proteins are involved in transcriptional repression, and PcG proteins are associated with transcriptional activation even more, acting independently of each other.

For instance, the chromatin remodeling factors BRM, SYD, PKL, and INO80 have an opposite effect to TrxG, facilitating the maintenance of a PcG-repressive chromatin state in a subset of genes [57,63,119,138]. It is noteworthy that one of the dual roles of PKL is through its interaction with RETINOBLASTOMA-RELATED 1 (RBR1) to repress gene expression [139]. Although the interaction of RBR1 and PcG proteins is well known [140], it is still unknown if this regulation by PKL includes PcG marks.

Besides chromatin remodeling factors, ULT1 has also the ability to physically associate with EMBRYONIC FLOWER1 (EMF1) a PcG protein, to maintain a genomic state of repression and chromatin integrity during germination; additionally, ATX1 also takes part in this process, and the *atx1 emf1 ult1* triple mutant surprisingly has fewer H3K27me3 levels than the *emf1* single mutant [141]. Indeed, it has been proposed that ULT1 in association with ATX1 forms a complex with PcG to maintain H3K27me3 repressive marks [141].

Furthermore, the H3K4me, in its mono-, di-, and trimethylation forms, are marks associated with transcriptionally active genes and this is highly conserved in eukaryotes [51,142]. However, recent genome-wide studies regarding the distribution and correlation of H3K4me2 with transcription in *Oryza sativa* revealed that at least in this species it could be acting as a repressive mark [143]. Additionally, in Arabidopsis, the H3K4me2 has also been found to colocalize at a higher-than-expected rate with the H3K27me3 PcG mark, which is associated with transcriptional repression [51]; however, its role in repression processes is unknown.

We can also point out the forcefulness of the effects that both TrxG and PcG marks have in the chromatin state and, consequently, in gene expression. Recent studies related to TrxG, specifically for ATX1, reveal that of the total number of downregulated genes in the mutant: very few presented a loss of the H3K4me3 mark [144]. In this study, in the *atx1* mutant, 3221 genes were downregulated, and of those, only 107 (3.32%) presented a loss of the H3K4me3 mark, breaking with the idea of a direct relationship between the absence of the mark and the dysregulation of gene expression, even if this dysregulation is in the expected sense. Further, the H3K27me3 and H2AK121ub PcG marks normally have an impact on the chromatin state and its responsiveness; however, the loss of function of PcG proteins and the consequent opening of chromatin is not necessarily the final effect to change gene expression [145].

Regarding TrxG’s relationship with Polycomb, sometimes, the understanding of the mechanisms in which TrxG regulates gene expression could be hindered due to the fixation in searching for its correlation with PcG. However, there are plenty of cases in which one can act independently from the other; of course, there are shared targets that fit the canonical model of antagonism between both, but on the other hand, there is a copious number of mutually exclusive targets. For example, it has been observed that the DREAM complex, a crucial regulator of cell proliferation that contains gene repression activity, can counteract TrxG function, regulating antagonistically a common set of genes [146,147], a function that is due to the interaction between BTE1 (a DREAM plant-exclusive subunit) and WDR5a, which inhibits the activity of the latter, and hinders the deposition of H3K4me3, thus avoiding efficient mRNA elongation at a subset of targets [146].

Therefore, it is not risky to suggest that the Polycomb–Trithorax regulatory system is highly dynamic, acting differentially between developmental stages, tissues, and even in specific genes.

## 9. Conclusions and Perspectives

The information concerning the TrxG in animals, as well as the information on the enzymatic activity, interactions, and function of each of the TrxG proteins described here, led us to gain insight into the mechanism of the TrxG complex in plants, including recruitment, the regulation of transcription, complexes, and roles in plant development and plasticity.

Despite all the information available, there is still a long way to understand each of the steps that involve this mechanism of gene regulation in plants. Although there is increasing information on the roles of specific plant TrxG components, more studies in conjunction but not individually are needed to gain insight into the roles of each component within the TrxG complex, likewise, to know the existence of the formation of other possible TrxG complexes that participate in specific processes of development, similar to those described for PcG [14].

Most of the information related to the TrxG mechanism in plants comes from studies at specific targets, mainly at the *FLC* locus. The structure of the *FLC* gene includes a large intron that, together with the lncRNAs encoded from different regions of the gene, play essential roles in its regulation by PcG and TrxG. This conserved structure is shared with other homeotic genes of the same family, a characteristic that could predict similar TrxG recruitment and transcriptional regulation mechanisms in this subset of genes. However, other TrxG targets with different gene structures or those that are specific targets for TrxG and not PcG could be used for the study of different mechanisms of recruitment and expression regulation. Therefore, further research is needed to identify the TrxG regulation mechanisms for all the different TrxG targets. Moreover, given the TrxG role in regulating plant plastic responses, it will be important to know if TrxG uses similar mechanisms to regulate intrinsic developmental processes and plastic responses.

Historically the research of TrxG has been carried out in comparison to its counteracting role with PcG. However, there is increasing evidence for the synergistic roles with the PcG of some TrxG proteins, in this regard, if both PcG and TrxG are able to regulate similar targets, it is not unreasonable to think that some members of both complexes could assist the opposite complex in specific cases. On the other hand, it will be important to approach the research in this field with a new perspective focusing on the TrxG mechanisms on targets that are not shared with PcG.

Finally, further research will be necessary to elucidate the TrxG roles in specific tissues or cells. For instance, the different cell types that conform to the plant stem cell niches have different lineages that transit from an undifferentiated to a differentiated state, and in this sense, defining which TrxG complex acts in each of these transitions and its regulation mechanisms would provide information regarding stem cell niche maintenance and plasticity.

## Figures and Tables

**Figure 1 plants-12-00075-f001:**
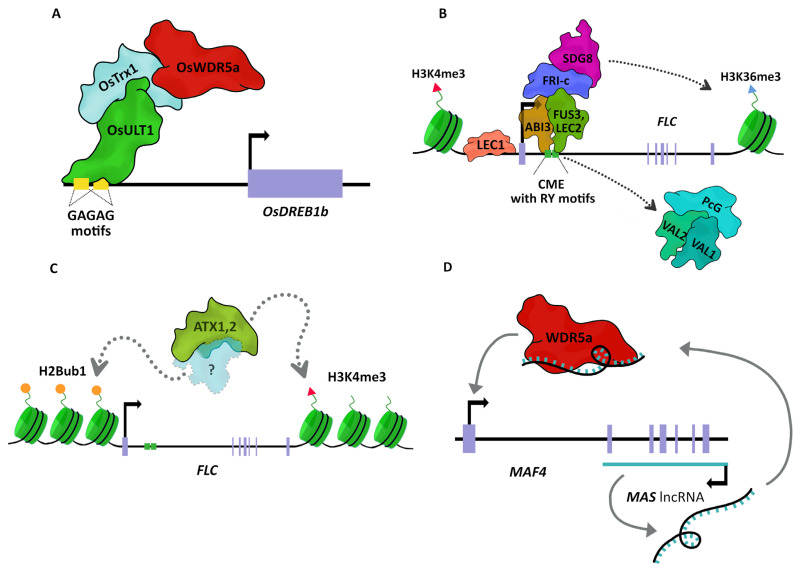
Different forms of TrxG recruitment in plants. (**A**) OsULT1 recognizes GAGA motifs on the OsDREB1b promoter to recruit the OsTrx1 along with the COMPASS protein OsWDR5a to activate transcription. (**B**) The B3-domain transcription factors ABI3, FUS3, and LEC2 recognize the RY motifs on *FLC* large intron, displacing PcG factors (including VAL1 and VAL2) (illustrated by an arrow) and recruiting FRI-complexed (FRI-c) with TrxG-SDG8 proteins to activate *FLC* transcription through the H3K36me3 deposition (indicated by an arrow). In this model, LEC1 binding increases chromatin accessibility into CME. (**C**) The H2Bub1 histone modification is recognized by an unknown protein (indicated by an arrow) that interacts with ATX1,2 to regulate *FLC* expression through the deposition of the H3K4me3 mark (indicated by an arrow). (**D**) The *MAS* antisense lncRNA transcribed from the *MAF4* locus interacts with WDR5a and may function as a guide to direct the recruitment of WDR5a at its target (interactions illustrated by arrows). CME: cold memory element.

**Figure 2 plants-12-00075-f002:**
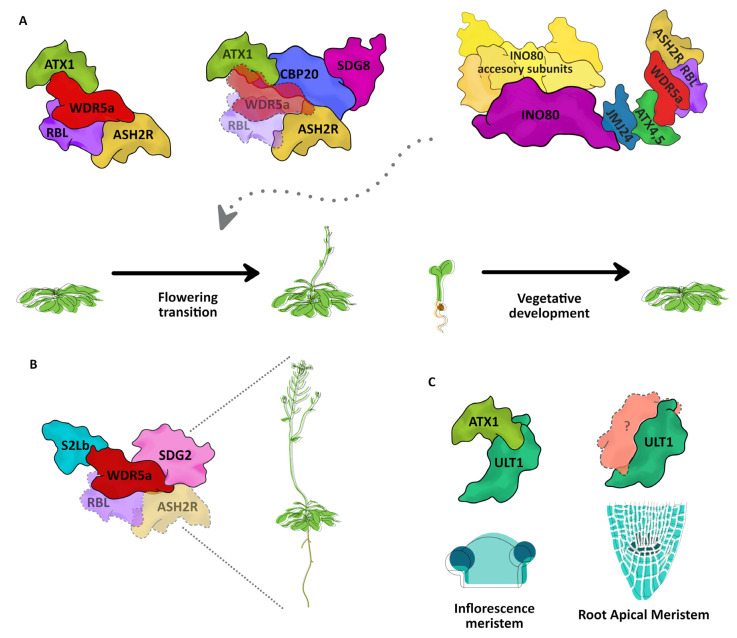
TrxG complexes reported in Arabidopsis. (**A**) The COMPASS-like complex formed by the core proteins WDR5a, RBL, and ASH2R interacts with ATX1 to control the flowering transition. SDG8 forms a complex with CBP20 which also interacts with ATX1 and ASH2R to regulate flowering transition. The chromatin remodeler complex INO80 interacts with the histone demethylase JMJ24, which in turn interacts with the ATX4 or ATX5 COMPASS-like complex to regulate the vegetative growth and the flowering transition (indicated by a dotted arrow). (**B**) The complex formed by the COMPASS protein WDR5a, which interacts with S2Lb and SDG2, regulates multiple processes in development, including root, vegetative, and reproductive processes. (**C**) ATX1 and ULT1 interact to form a complex that regulates the differentiation of inflorescence meristem, whereas, in the root, ULT1 regulates the maintenance of the root apical meristem independently of ATX1. Dotted proteins represent predicted interactions based on experimental data.

**Figure 3 plants-12-00075-f003:**
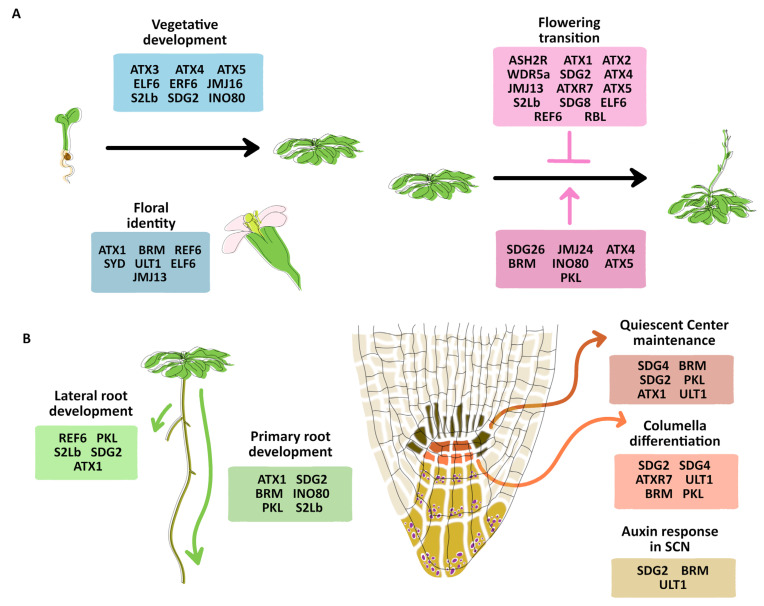
Roles of TrxG proteins in development. (**A**) List of TrxG proteins involved in the vegetative development, floral identity, and flowering transition. The pink arrow indicates flowering promotion, whereas the flat pink arrow indicates a flowering restriction. (**B**) List of TrxG proteins involved in the lateral and primary root development, including the quiescent center maintenance, columella differentiation, and auxin response on the SCN. SCN: Stem cell niche.

**Table 1 plants-12-00075-t001:** List of Arabidopsis TrxG components. TrxG proteins are listed by their classification, showing molecular function, domain associated, as well as their homologs in rice, *Drosophila melanogaster* (flies), and mammals.

Classification	Plants		Flies	Mammals	Function	Domain
	Arabidopsis	Rice				
**Histone methyltransferases (HMTs)**	ATX1/SDG27 [16] ATX2/SDG30 [17] ATX3/SDG14 [18] ATX4/SDG16 ATX5/SDG29 [18]	OsTrx1 /SDG723 [19] OSDG721 [20] OSDG705 [20]	Trx Trr	MLL1/2 MLL 3/4	H3K4 HMT H3K4 HMT H3K4 HMT	SET SET SET
	ATXR2/SDG36 [21] ATXR3/SDG2 [22]	OSDG701 [23]			H3K4/36, HMT H3K4 HMT	SET SET
	ATXR7/SDG25 [24,25]		Set1	SET1 A/B	H3K36/4, HMT	SET
	ASHH2/SDG8/EFS [26]	OsSDG724 and OsSDG725 [27,28]	Ash1	ASH1L	H3K36 HMT	SET
	ASHH1/SDG26 [29]	OsSDG708 [30]			H3K36/4, HMT	SET
	ASHR3/SDG4 [31]				H3K36/4, HMT	SET
**COMPASS** **Core**	WDR5a [32]	OsWDR5a [33]	Wds	WDR5	Histone binding	WD40
	ASH2R/TRO [34]	OsASHL1 OsASHL2 OsASHL3 [35]	Ash2	ASH2L	DNA binding	Zinc Finger
	RBL [34]	OsRBL [35]	Rbbp5	RBBP5	Histone binding	WD40
	S2Lb [36]		-	WDR82	Histone binding	WD40
**Chromatin Remodelers**	SWI/SNF: BRM [37]		BAP	BRM/BRG1	ATP-dependent chromatin remodeler	Bromodomain
	SYD [38]				ATP-dependent chromatin remodeler	Bromodomain
	INO80 [39]		INO80	INO80	ATP-dependent chromatin remodeler	DBINO
	CHD3: PKL [40]	OsCHR729 [41]	CHD3	CHD3	ATP-dependent chromatin remodeler	CHD
**Histone Demethylases**	ELF6 REF6 JMJ13 [42]	OsJMJ705 [43]	Utx	JmjC proteins	H3K27 demethylase	jmjC
	JMJ24 JMJ26 JMJ28 [39]	OsJMJ706 [44]			H3K9 demethylase	jmjC
**Co-activators**	ULT1/2	OsULT1 [45]	-	-	ATX1 coactivator	SAND
	SEC		Esc (Polycomb gene)	OGT	ATX1 coactivator	O-linked N-acetylglucosamine transferase

## Data Availability

Not applicable.

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
