# Peer review of "A Green Light to Switch on Genes: Revisiting Trithorax on Plants"

_plants, 2022, doi:10.3390/plants12010075_

Round 1

Reviewer 1 Report

In this review the authors aim to discuss the Trithorax group complexes in plants, their role in gene activation and in plant development. It focuses on different classes of Trithorax group proteins and their complexes and how they are recruited to the target sites. Further the authors discuss the role of Trithorax proteins in plant development, specifically root and flower development. The review addresses an important subject which is Trx proteins/complexes in plants and its role in plant development. It is well written and structured. Please see my recommendations below.

-The overall resolution of the figures should be improved, the entire review will benefit from this.

-I suggest including a table with names of different TrxG genes and their homologs in different plants such as Arabidopsis, rice etc.

-Finally, I recommend writing a section on Trithorax involvement in leaf development in plants.

Author Response

Thanks for the recommendations to improve our manuscript

-The overall resolution of the figures should be improved, the entire review will benefit from this.

Reply:  According to this suggestion, in the new version we included new figures with higher resolution (600dpi) and we also increased the size of legends and the color of some protein names to give more contrast

-I suggest including a table with names of different TrxG genes and their homologs in different plants such as Arabidopsis, rice, etc.

Reply:  As there is more information on the function of TrxG members in rice, we included in table 1 a column with genes described with function related to TrxG 

-Finally, I recommend writing a section on Trithorax involvement in leaf development in plants.

Reply: Although it is a very good suggestion, especially considering whether genes involved in leaf formation and development such as KNOXs, which are PcG targets, are also regulated by TrxG. However, in our knowledge, there is insufficient information on the role of TrxG in leaf development. Although the characterization of some TrxG mutants show changes in leaf size or morphology, the role of TrxG proteins in the regulation of genes related to leaf formation, morphology, growth, etc. is still unknown.

Reviewer 2 Report

This is a well-written review on an important topic. I noticed only one minor thing that should be corrected. In the Table 1 legend on line 81, 'homologous' should be 'homologs'. 

Author Response

This is a well-written review on an important topic. I noticed only one minor thing that should be corrected. In the Table 1 legend on line 81, 'homologous' should be 'homologs'.

Reply: Thank you, in the new version we corrected this error and other text errors.

Reviewer 3 Report

This article provides an updated and comprehensive review about the interactions and roles of TrxG proteins in plants. The TrxG mechanism to regulate transcription, the TrxG complexes formed as well as their roles during plant development and plasticity are discussed. The manuscript is well written. It provides an interesting and timely contribution for the functional interaction of TrxG components, which will stimulate further studies on related research topics in plants.

Author Response

Thanks for your comments